# Research on Real Purchasing Behavior Analysis of Electric Cars in Beijing Based on Structural Equation Modeling and Multinomial Logit Model

**Qingyou Yan [1,2], Guangyu Qin [1,2,*], Meijuan Zhang [1,2] and Bowen Xiao [3]**

[1]  School of Economic & Management, North China Electric Power University, Beijing 102206, China;
    yanqingyou@ncepu.edu.cn (Q.Y.); zhangmeijuan@ncepu.edu.cn (M.Z.)
[2]  Beijing Key Laboratory of New Energy & Low Carbon Development, North China Electric Power University,
    Beijing 102206, China
[3]  School of Economics and Management, Beihang University, Xueyuan Road, Haidian District,
    Beijing 100191, China; xbw468@ncepu.edu.cn
*   Correspondence: qinguangyu@ncepu.edu.cn

**Abstract:** At present, electric cars are being developed rapidly in China as emerging carbon emission reduction vehicles, but their proportion in the Chinese automobile market is still small, and a large number of potential consumers are still holding a wait-and-see attitude. Therefore, for the sake of promoting the further development of electric cars in China, this paper based on the TPB (Theory of Planned Behavior) theoretical research framework, investigates potential consumers in typical areas of Beijing and collects a large amount of data through the design of paper and electronic questionnaires. SEM (Structural Equation Modeling) and MNL (Multinomial Logit Model) models are used to analyze key factors affecting consumers' purchase intention and actual purchasing behavior. The results show that the positive and negative attributes of consumers' attitude, subjective norm, and perceived behavior control will have different effects on consumers' actual purchasing behavior. Finally, based on the analysis results, some reasonable suggestions are proposed for the government and EV (Electric Vehicles) enterprise service providers to increase electric vehicle diffusion.

**Keywords:** electric cars; theory of planned behavior; structural equation modeling; multinomial logit model; purchase intention; real purchasing behavior

---

## 1. Introduction

Currently, the environmental quality of many Chinese cities is deteriorating year by year. Air pollution caused by fog and haze, in particular, is becoming increasingly severe. The air pollution levels in China are already above the "safe level" [1] set by the European Union, and the primary cause is the rapid growth in the purchase and use of gasoline cars and the smaller market share of the electric cars [2]. Enough attention has also been paid to the seriousness of environmental problems. For example, the government has also introduced some policies [3], including purchase subsidies [4] and tax exemption policies [5], to encourage consumers to buy electric cars, so as to improve consumers' environmental awareness and promote their willingness to buy electric cars.

In spite of the continuous fast growth in the number of electric cars with each passing year, electric cars still account for only a small proportion of total cars. According to the latest data, China sold 1.25 million electric cars in 2018, accounting for 4.4% of new car sales, up just 1.7% from last year [6]. This data shows that the popularity of electric cars in China still has a long way to go; most consumers are in a state of wait and see [7]. In order to acquire driving factors and barriers, former researchers have investigated a number of factors using different methods

ranging from revealed preference to stated preference [8–15]. Safety aspects of electric cars, such as risk due to low noise and reliability of the cars, were surveyed in some previous studies [16–18]. Takanori Okada et al. used SEM(Structural Equation Modeling) to analyze the purchase intention of users who do not own electric cars and the purchase satisfaction of users who already own electric cars, and proposed that environmental awareness will have a direct impact on the purchase intention of users who do not own an EV(electric vehicles), while on the contrary, it will have an indirect impact on the purchase satisfaction of users who already own an EV. It subdivides consumers and makes a differential analysis of the role of environmental awareness. A clear distinction to the relation by using SEM makes it more logical [19]. Dooyoung Choi et al. extended the TPB (Theory of Planned Behavior) theory and found that only attitude and subjective norm can satisfactorily influence the purchasing tendency of green products. It empirically explored the dimensions and significance of the predictive variables. But possible selection biases and no actual purchasing behavior measurement would limit its generalizability [20]. Liu Han et al. proposed that functional values of electric cars, such as convenience, performance, and monetary, have a direct and indirect effect on the purchase tendency of electric cars, while non-functional values, such as emotional, social responsibility, and social identity, has an indirect effect on the purchase tendency of electric cars under the adjustment of attitude. Performance values, such as reliability, endurance, and charging time, have the greatest impact on the purchase tendency of electric cars. Such distinction adds to the pertinence and accuracy of research [21]. Ozlem Simsekoglu et al. put forward that subjective norm, perceived behavioral control, and perceived attributes of electric cars have positive effects on purchase intention and that gender has different influences on consumer's purchase intention. However, the problem of low sample data may limit its generalization [22]. Xiuhong He et al. explored consumers' purchase intention of electric cars by using the research framework of personality-perception-intention, and introduced that in the action path of personality on purchase intention, positive utility and negative utility of perception will have different mediating effects on its action mode [23]. Nonetheless, because the previous researches mostly used an online survey platform to collect data [22,23], it may result in sample bias, therefore the obtained data might have lower credibility and restricted universality. Moreover, most of these studies did not take into account the real purchasing behavior of electric cars.

In order to promote the further development of electric cars in China, this paper explores the influencing factors of the actual purchasing behavior of electric cars. By using the SEM and MNL(Multinomial Logit Model) models, this paper studies the positive and negative attributes of the attitude, subjective norm, and perceived behavior control affecting the actual purchasing behavior of electric cars. The findings could provide guidance for the government and suppliers to take countermeasures. We designed paper and online questionnaires to enhance the response rate, interviewed typically conventional drivers, and collected a large number of questionnaire survey data. Furthermore, based on the theory of planned behavior, this paper builds the SEM and MNL model to explore the positive and negative factors affecting the intention, and in turn, the real purchasing behavior of electric cars has also been studied. Finally, through the analysis of various factors, proposal is made to support the decision of the government and related electric vehicle service providers by finding out the key factors affecting consumers when buying electric cars.

### 1.1. Theory of Planned Behavior

Several studies have examined the role of social and psychological factors for adoption of electric cars using the theory of planned behavior as a theoretical framework [24–27]. The theory of planned behavior is developed from the theory of reasoned action, which is based on the theory of expectancy value, explaining individual decision-making process from the perspective of psychology [28]. It predicts and understands human behavior by weighing the potential determinants' behavior: behavior intention is a process of accumulation and reinforcement of thought tendency and motivation. The stronger the intention is, the more likely the action [29,30]. The behavior attitude, subjective norms, and perceived behavior control jointly determine individual

intention (Figure 1). Individual characteristics, attitudes and beliefs about things, work characteristics, environment, and other external factors determine behavior attitudes, subjective norms, and perceived behavior control.

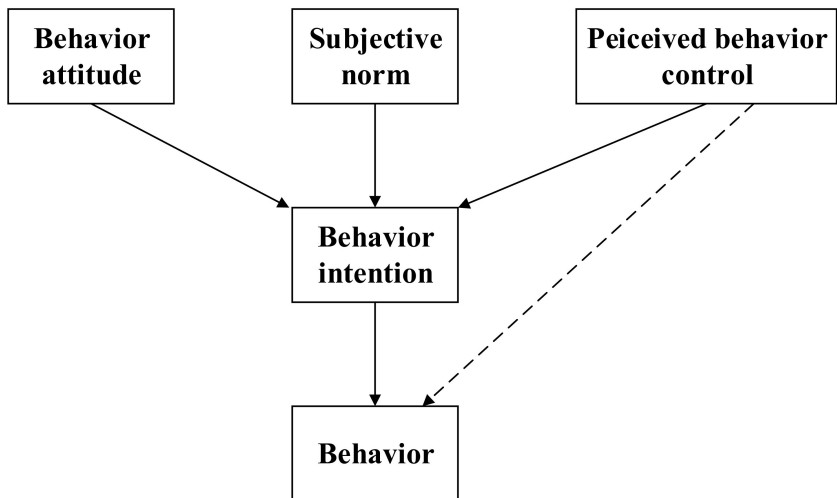

**Figure 1.** Planned behavior theory model [24,30].

### 1.2. Behavior Attitude Related to Electric Cars

Behavior attitudes represent potential consumers' assessment of the positive or negative impact of buying electric cars [31]. When potential consumers believe that electric cars are environment-friendly, cost-efficient, and supported by government policies, produce less noise, and could meet daily requirements, then they will have a positive attitude toward buying electric cars. On the contrary, potential consumers will have a negative attitude toward buying electric cars when they believe that it is inconvenient to charge electric cars, that electric cars themselves often have problems, and that their cruise range cannot meet their daily needs.

### 1.3. The Subjective Norm Related to Electric Cars

Subjective norms represent the degree to which an individual perceives important people to expect an individual to perform a certain behavior [32]. In combination with China's national conditions, due to the dominance of the government and the high-power distance characteristics of the Chinese society, potential consumers generally show respect and obedience to behaviors advocated by the government. For example, government subsidies and tax reduction policies have a strong "demonstration" effect on potential consumers. Friends, family, media publicity, electric vehicle enterprise service providers, and the construction of "vehicle-pile-network" platform are important social network resources for potential consumers. Before making decisions, potential consumers often seek opinions from friends and relatives, evaluating whether the corresponding service resources are compatible, and finally make their decisions.

### 1.4. The Perceived Behavior Control Related to Electric Cars

Perceived behavior control represents the individual's perception of the difficulty of performing a certain behavior [29]. The more resources and opportunities the individual thinks he has, the stronger the perceived behavior control will be. When potential consumers have sufficient economic capacity and have the right to decide and there are sufficient charging resources around their living and working areas, potential consumers will think it is easy to buy an electric vehicle in the future and have stronger control over the perceived behavior of buying electric cars.

### *1.5. Demographic and Socio-Economic Characteristics*

Although the paper does not focus on the influence of demographic characteristics on purchasing behavior of electric cars, we still add three main control variables into the model. Gender, monthly income, and educational background affect the purchasing behavior of electric cars, so it is feasible to add them into the model.

### *1.6. Aims of the Study*

The main purpose of the current study is to explore the key factors influencing potential customers' actual purchase of electric cars, so as to provide reasonable policy suggestions for the government and electric enterprise service providers. Based on the planning behavior theory, this paper constructs a framework that affects the actual purchasing behavior of electric cars (Figure 2). Statistical data of traditional drivers aged 30–60 years in the Future Science City of Changping District of Beijing and North China Electric Power University were collected by issuing paper questionnaires and online questionnaires, then the structural equation model (SEM) and the multinomial logit model (MNL) were combined to process and analyze the data. Finally, the key factors that affect potential consumers' actual purchase of electric cars were concluded, thus providing reasonable suggestions for the government and EV enterprise service providers.

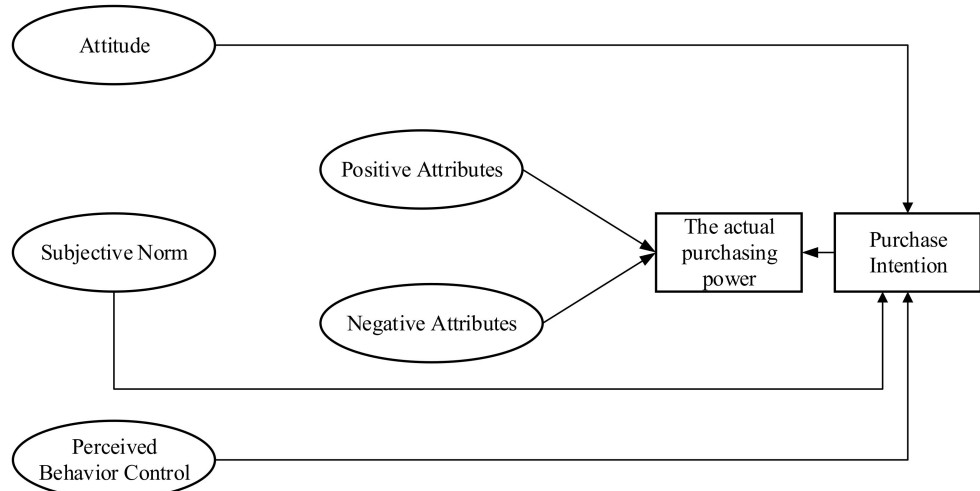

**Figure 2.** The framework of the actual purchasing behavior of electric cars.

The rest of the paper is organized as follows. Section 2 will present the theoretical framework and data collection of the paper. Then, we conduct model analysis on the data collected from the questionnaire and discuss the results in Section 3. Finally, the conclusion of the paper will be presented in Section 4. This paper makes up for deficiency in previous research, mainly from three aspects of improvement. Firstly, SEM and MNL models are combined to conduct the case analysis by means of on-site and online questionnaires. Secondly, the purchasing intention is further concretized into real purchasing behavior to provide more favorable guidance for the diffusion policy of electric cars. Thirdly, the TPB theoretical framework will be further developed to analyze the positive and negative attributes of attitude, subjective norm, and perceived behavior control and explore the main and key factors influencing purchase intention and real purchasing behavior.

## 2. Methods

### *2.1. Theoretical Framework and Hypothesis Development*

Combined with previous studies and the TPB theory, we first give the following hypothesis.

**H1:** *The more positive the consumers' attitude toward electric cars is, the stronger their purchase intention of electric cars and actual purchasing behavior;*

**H1:** *The subjective norms of consumers are positively related to the purchase intention of electric cars, that is, the stronger their subjective norms are, the stronger their willingness to purchase electric cars and actual purchasing behavior;*

**H3:** *The consumers' perceived behavioral control is positively related to the purchase intention of electric cars, that is, the stronger the perceived behavioral control is, the stronger their willingness to purchase electric cars and actual purchasing behavior.*

This paper will use SEM and MNL models to test the above assumptions [19,20,23,33–37]. On the basis of verifying the influence path of attitude, subjective norm, and perceived behavior control on purchase intention, we will use the MNL model to further study the impact of positive and negative attributes as intermediate variables on actual purchasing behavior. The research framework is shown in Figure 3.

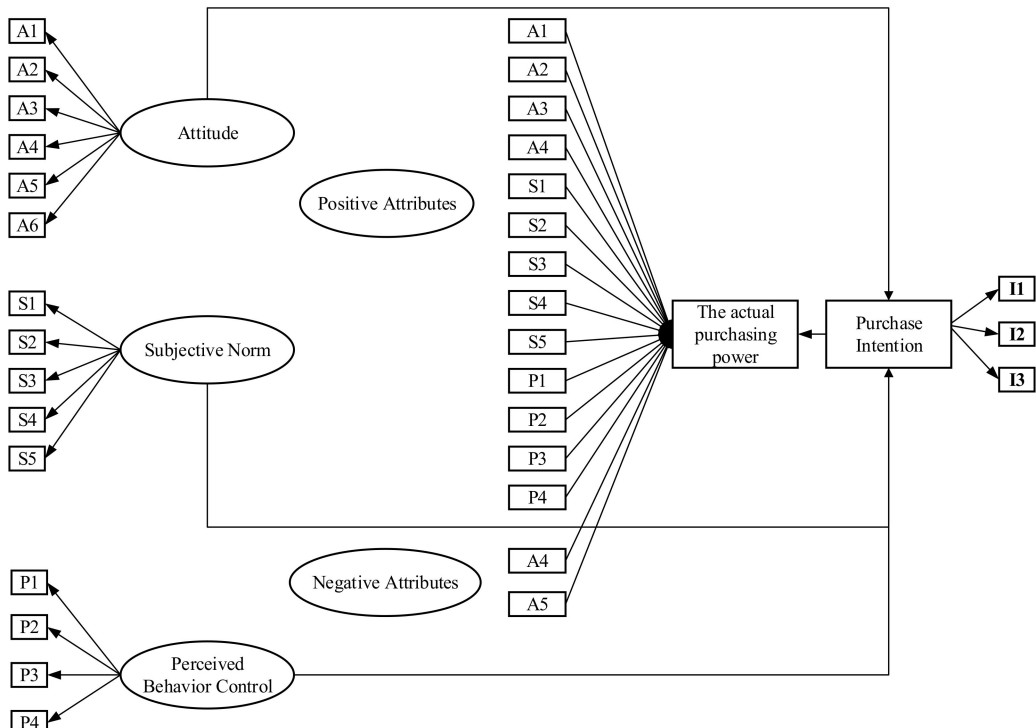

**Figure 3.** The SEM-MNL (Structural Equation Modeling and Multinomial Logit Model) of the factors influencing the purchase intention of electric cars.

*2.2. Data Collection, and Demographic and Socio-Economic Characteristics Analysis*

This paper conducted a questionnaire survey on the Future Science City of Changping District and North China Electric Power University in Beijing from 1 July 2019 to 1 August 2019. The Future Science City takes the low-carbon energy conservation as the development principle and is the national key innovation and technology park, with early investment in electric cars. At the same time, the North China Electric Power University, relying on the characteristics of electricity, has also invested in electric cars earlier. The users have a better understanding of electric cars, and relevant infrastructure and policies have been implemented relatively well. At present, the infrastructure construction in other regions is not perfect, the vehicle networking platform is not built, and the knowledge popularization of electric cars is low, which leads to the lack of scientific sample collection. As the government vigorously promotes energy conservation and emission reduction, environment-friendly characteristics of electric

cars are increasingly well known, and the consumers will gradually increase their understanding of electric cars. Consumers in this region will represent the future of people's general cognitive condition of electric cars, so considering the time lag, its findings will be certainly prospective and representative.

The five-point Likert scale was used in the questionnaire: 5, 4, 3, 2, and 1 respectively represent "strongly agree", "agree", "uncertain", "disagree", and "strongly disagree". The contents used for measurement were designed on the basis of previous studies [38–44]. On the strength of the latest research, we expand the observation variables of subjective norm and perceived behavior control, which are not limited to previous questionnaires. Meanwhile, we set the purchase intention on a hierarchical basis, which represents the different intensity of consumers' purchase intention and link up with the subsequent analysis of actual purchasing behavior. Before the formal investigation, we conducted a pilot project within the North China Electric Power University, and analyzed 65 valid questionnaires collected, and revised some of the topics in combination with the analysis results.

The questionnaire uses a combination of online survey and on-site paper questionnaire distribution. See Appendices A and B for the specific contents of the questionnaire. We focused our survey on the staff of North China Electric Power University and the Future Science City, and the age range was 30–60 years old. We believe that this group has a strong conceptual awareness of electric vehicles and a deep degree of cognition, and the region has a complete infrastructure and early investment, so it is representative and can be deduced from the part to the whole. According to the number of people in this area, about 1000 questionnaires were issued, and a total of 621 questionnaires were recovered. Finally, a total of 537 valid questionnaires were screened out. As for sample selection, we set up a selection mechanism to screen out 537 valid questionnaires from a total of 621 questionnaires. We investigated the quality of the questionnaire from the aspects of vacancy rate, completeness, filling time, and regular response, and eliminated the questionnaires that did not meet the requirements. The overall analysis of the questionnaire shows that the scale of the questionnaire is quite relevant. Its demographic characteristics are shown in Table 1. The results show that around 60% of respondents are male, the majority of whom have bachelor's or graduate degrees. Respondents with a monthly income of 5000–15,000 are the majority. Our survey is mainly aimed at consumers in their prime age as drivers, that is, 30–60 years old. The analysis found that the majority of respondents were aged between 30 and 50. We take the above four measures as the control variables for analysis, and the results show that the slope coefficients of the three measures for the purchase intention are not significant, for this reason their influence could not be proved.

**Table 1.** Demographic characteristics.

| Measure | Item | Count | Percentage |
|---|---|---|---|
| Gender | Male | 322 | 59.96% |
| | Female | 215 | 40.04% |
| Educational background | High school or below | 26 | 4.84% |
| | Bachelor's degree | 134 | 24.95% |
| | Master's degree | 287 | 53.45% |
| | Doctor degree or above | 116 | 21.60% |
| Monthly income | <5000 RMB | 71 | 13.22% |
| | 5000–8000 RMB | 112 | 20.86% |
| | 8000–12,000 RMB | 145 | 27.00% |
| | 12,000–15,000 RMB | 97 | 18.06% |
| | 15,000–20,000 RMB | 60 | 11.17% |
| | >20,000 RMB | 52 | 9.68% |
| Age | 30–40 | 211 | 39.29% |
| | 40–50 | 179 | 33.33% |
| | 50–60 | 147 | 27.38% |

RMB (RenMinBi Yuan).

## 3. Data Analysis and Discussion

### 3.1. Measurement Model

The reliability and validity of the measurement model are tested using SPSS and Amos software. As shown in Table 2, the Cronbach's alpha value of each variable is above 0.6, indicating that the model passes the reliability test. Meanwhile, the loadings of all items are all greater than 0.7, implying that items are highly representative of their latent variables. In addition, the average variance extracted (AVE) is greater than 0.5 and the composite reliability (CR) value of each construct is greater than 0.8, suggesting that the model has good convergence validity. In addition, as shown in Table 3, the correlation coefficient between attitude, subjective norm, perceived behavior control, and purchase intention demonstrates a significant correlation with each other ($p < 0.01$). At the same time, their absolute values are all less than 0.5 and all of them are less than the square root of AVE, which manifestly proves that all variables have certain correlation and definite discrimination, and the discriminant validity of the data is relatively good.

Besides, this paper tested the overall fitness of the model. The calculation results show that the ratio of Chi-square to degree of freedom is 2.031, whose value is less than 3, revealing that the fitness is acceptable. In the meantime, other indicators, such as RMSEA(Root Mean Square Error of Approximation), AGFI(Adjusted Goodness of Fit Index), CFI(Comparative Fit Index), and TLI(Tucker-Lewis Index), are 0.041, 0.927, 0.973, and 0.984, respectively, which are all within the receivable range, indicating that the model has good structural validity and its overall fitting is great.

**Table 2.** Confirmatory factor analysis test (CFA) results.

| Path | Loadings | AVE | Composite Reliability | Cronbach's Alpha Value |
|---|---|---|---|---|
| Attitude | | | | |
| A1 | 0.911 | | | |
| A2 | 0.937 | | | |
| A3 | 0.846 | 0.728 | 0.930 | 0.874 |
| A4 | 0.913 | | | |
| A5 | 0.798 | | | |
| A6 | 0.887 | | | |
| Subjective Norm | | | | |
| S1 | 0.821 | | | |
| S2 | 0.783 | | | |
| S3 | 0.862 | 0.687 | 0.893 | 0.905 |
| S4 | 0.797 | | | |
| S5 | 0.882 | | | |
| Perceived Behavior Control | | | | |
| P1 | 0.835 | | | |
| P2 | 0.919 | 0.836 | 0.924 | 0.931 |
| P3 | 0.924 | | | |
| P4 | 0.902 | | | |
| Purchase Intention | | | | |
| I1 | 0.716 | | | |
| I2 | 0.811 | 0.760 | 0.919 | 0.859 |
| I3 | 0.875 | | | |

**Table 3.** Discriminant validity test results. Legend: AVE (Average Variance Extracted)

| Variable | Attitude | Subjective Norm | Perceived Behavior Control | Purchase Intention |
|---|---|---|---|---|
| Attitude | 0.728 | | | |
| Subjective Norm | 0.324 ** | 0.687 | | |
| Perceived Behavior Control | 0.339 ** | 0.360 ** | 0.836 | |
| Purchase Intention | 0.258 ** | 0.284 ** | 0.290 ** | 0.760 |
| The square root of AVE | 0.853 | 0.829 | 0.914 | 0.872 |

Note: *** $p < 0.001$; ** $p < 0.01$; * $p < 0.05$.

### 3.2. Research Model

Next, we tested the model hypothesis. The test results are shown in Figure 4. Firstly, in SEM framework, the path coefficients of the attitude, subjective norm, and perceived behavior control for purchase intention are 0.037, 0.326, and 0.270, respectively, and both are significant at the 1% level. This indicates that consumers' attitude toward electric cars, subjective norm, and perceived behavioral control has a prominent effect on their purchase intention, supporting the aforementioned three assumptions. However, consumers' attitude toward electric cars has little influence on purchase intention, which may be due to the differential impact of positive or negative attributes of variables on purchase intention.

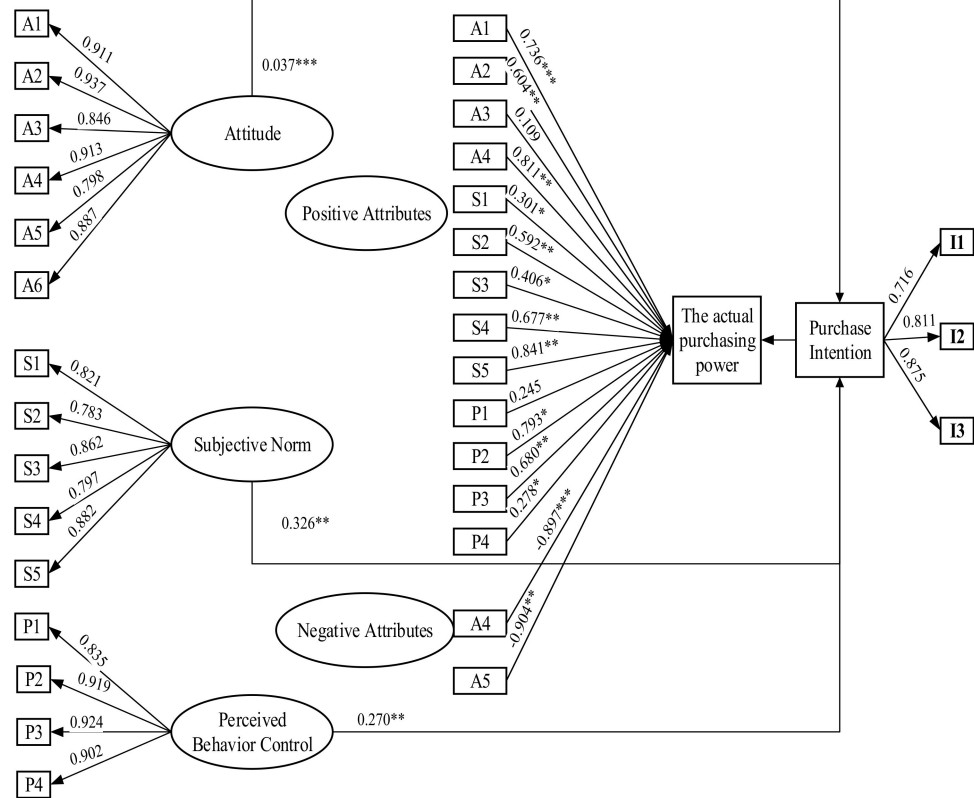

**Figure 4.** The results of the research model. (Note: *** $p < 0.001$; ** $p < 0.01$; * $p < 0.05$).

In order to further understand the impact of the positive and negative attributes of each variable as intermediate variables on the actual purchasing behavior, this paper distinguishes each observed variable and assumes that I1, I2, and I3 mean the actual purchasing behavior of the probability of 30%, 60%, and 100%, respectively. The three items are weighted and their scores ranged from 1.9 to 9.5. The sequence of classification variables representing the actual purchasing behavior of each

sub-interval is defined as 1, 2, 3, 4, and 5, respectively, after five-equal division of this interval, that is, the score in the sub-interval (1.9–3.8) corresponds to the point 1 of the sequence, and so on. In the above process, the three observed variables that reflect purchase intention are integrated according to different probabilities, so as to maximize the use of observation information to express consumers' actual purchasing behavior and facilitate the connection between purchase intention and actual purchasing behavior, thus making it easier to carry out the follow-up research.

On the basis of the above results, we reconstructed the data by establishing the multinomial logit model and used Eviews software to analyze whether the positive and negative attributes of each variable affect the actual purchasing behavior. The multinomial logit model is built on the strength of random utility theory [45–48]. When a rational decision-maker is faced with a choice, he will make the choice according to the principle of utility maximization. The explanatory variables of the model include all the factors that affect the selection, including the attributes of decision makers and that of alternatives. In this paper, we distinguish the observed variables into positive and negative attributes. Through the establishment of the multinomial logit model [49–55], we further study the influence of various attribute factors on the actual purchasing behavior of electric cars.

The model can be expressed as follows:

$$U_{\alpha\beta} =_{\alpha\beta} +\varepsilon_{\alpha\beta}. \tag{1}$$

The probability of choosing j is:

$$(U_{\alpha\beta} > U_{\alpha\gamma}), \gamma = 0, 1, 2, \ldots, \Lambda, \gamma \neq \beta. \tag{2}$$

While the random error term is independent and its distribution follows the extreme value distribution, the probability of choosing j can be calculated as Equation (3):

$$(\Psi_\alpha = \beta) = \frac{e^{\alpha\beta}}{\sum\limits_{\beta=0}^{\Lambda} e^{\alpha\beta}}. \tag{3}$$

The maximum likelihood method is used to estimate the model coefficient matrix *B* to study the influence degree of each characteristic factor on the purchasing behavior of electric cars [56]. The likelihood ratio tests show that each influencing factor has a certain influence on the explained variable. The regression results are shown in Table 4.

According to the regression results of the model, most of the explanatory variables were significant at the 5% level. We calculate the accuracy of the model by subtracting the percentage of opportunity accuracy (PCHAR) from the overall percentage of accurate predictions. The accuracy was 27% more than the proportional by chance accuracy rate (0.66 vs. 0.52). Meanwhile, combined with McFadden Pseudo $R^2$ and LR statistics, the model fitted well, showing that various attributes of attitude, subjective norm, and perceived behavior control have remarkable influence on actual purchasing behavior.

In terms of positive attributes, the low pollution emissions, low cost, and government-related support policies of electric cars have a significant impact on the actual purchasing behavior of electric cars, and their impact is positive. The three attributes are significant at the level of 1%, and the influence coefficient of A4 is 0.811, which is larger than the other two (0.736 and 0.604), indicating the most influential attribute. It confirms that on the path of influence of consumers' attitude on purchase intention, the positive attributes as a dominant intermediate variable will have a strong positive effect on purchasing intention and the actual purchasing behavior of electric cars. Compared with fuel cars, electric cars could produce less pollution. Besides, charging costs are relatively low compared with fuel costs and other expenses. In combination with government subsidies and welfare policies, consumers have a positive attitude toward purchasing electric cars, thus improving the actual purchasing behavior of electric cars.

**Table 4.** Estimated results of MNL model.

| Categories | Attributes | Coefficient | Standard Error | OR |
|---|---|---|---|---|
| | A1 | 0.736 *** | 0.031 | 2.088 |
| | A2 | 0.604 ** | 0.057 | 1.829 |
| | A3 | 0.109 | 0.452 | 1.115 |
| | A4 | 0.811 ** | 0.063 | 2.250 |
| | S1 | 0.301 * | 0.104 | 1.351 |
| | S2 | 0.592 ** | 0.089 | 1.808 |
| Positive Attributes | S3 | 0.406 * | 0.174 | 1.501 |
| | S4 | 0.677 ** | 0.131 | 1.968 |
| | S5 | 0.841 ** | 0.093 | 2.319 |
| | P1 | 0.245 | 0.375 | 1.278 |
| | P2 | 0.793 * | 0.118 | 2.210 |
| | P3 | 0.680 ** | 0.066 | 1.974 |
| | P4 | 0.278 * | 0.170 | 1.320 |
| Negative Attributes | A5 | −0.897 *** | 0.042 | 0.408 |
| | A6 | −0.904 ** | 0.080 | 0.405 |
| **Statistical Performance** | | | | |
| McFadden Pseudo R2 | | 0.759 | | |
| LR statistic | | 51.34 | | |
| Prob. (LR statistic) | | 0.00 | | |

Note: *** $p < 0.001$; ** $p < 0.01$; * $p < 0.05$.

Media publicity, government subsidy policies, high-quality supplier services, and the establishment of Internet of Cars platform have shown a noteworthy positive impact on the actual purchasing behavior of electric cars, consistent with the conclusion described above that subjective norms have a positive impact on purchase intention. The results show that the coefficient of S5 was the largest among them, which was 0.841, revealing that the effect is remarkable. At the beginning of 2019, China's Internet of Cars platform was basically completed. The Internet of Cars platform makes use of on-board electronic sensing devices to realize information interconnection through mobile communication technology, GPS(Global Position System), intelligent terminal equipment, and information network platform, so as to carry out effective intelligent monitoring, dispatching, and management network system for cars and roads, greatly promoting the safety and convenience of electric cars. Meanwhile, in early 2019, the State Grid Electric Vehicle Service Company Ltd. launched the "e-smart" intelligent electric socket, which changed the traditional public and private charging piles and realized convenient shared charging of users [57]. These positive traits attributed to subjective norms have a positively receivable effect working as intermediate variables on the actual purchasing behavior of electric cars. Under the influence of these positive characteristics, consumers have strong positive subjective norms, thus making consumers have a strong purchase intention.

These qualities, attributed to the positive characteristics of perceived behavioral control, including economic strength and charging resource support, have displayed a significant positive effect on the purchase intention of electric cars with values of 0.793 and 0.680, separately. The stronger the economic strength of consumers and the better the support of surrounding charging resources, more powerful is the consumers' perceived behavioral control, the higher their purchase intention, and the greater their actual purchasing behavior of electric cars.

In terms of negative attributes, the fixed charging facilities have a conspicuous negative effect on the actual purchasing behavior of electric cars, for the coefficient of −0.897 and −0.904 on the 1% significance level. The immobility nature of charging piles cannot meet the huge demand, nor can it cope with the urgent need of electricity during driving. Meanwhile, mobile charging piles have not been popularized yet, and consumers have little understanding of them. For that, consumers have

a negative attitude toward electric cars, which gives rise to a negative impact on the actual purchasing behavior of electric cars. The susceptibility to damage and poor endurance of electric cars bring great inconvenience to daily users, which leads to the negative attitude of consumers toward electric cars, resulting in a low propensity to purchase, thus having a negative effect on the actual purchasing behavior of electric cars. To some extent, the different impact of positive and negative attributes of attitude on purchase intention as intermediate variables can explain the low path coefficient of the aforementioned attitude on purchase intention.

According to the above analysis results, we first propose that the government should perfect the subsidy and relevant alternative incentive policy of electric cars to improve the real purchasing behavior of consumers on the premise of considering the budget. Next, manufacturers are supposed to speed up technological research and development, advance the performance of electric cars, reduce costs, provide high-quality supplier services, and publicize their excellent quality by media, so as to promote consumers' purchasing intention and actual purchasing behavior. Lastly, the construction of the Internet of Cars platform needs to be further improved to make up for the lack of actual operation of electric cars and make overall planning for the charging resources to augment the consumption potential of the electric vehicle market.

## 4. Conclusions

Based on the TPB theory, we analyzed the factors affecting the actual purchasing behavior of electric cars. Under the TPB theory framework, we first studied the effects of attitude, subjective norm, and perceived behavior control on purchase intention, and the path coefficients of the three are significant. On this basis, we discretize the three observed variables of purchasing intention as the actual purchasing behavior, and further study the impact of positive and negative attributes of the attitude, subjective norm, and perceived behavior control on the actual purchasing behavior.

The results show that positive attributes have a significant positive effect on actual purchasing behavior, while, conversely, negative attributes have an adverse impact. Firstly, the low pollution emissions, low cost, and government-related support policies of electric cars have made consumers have a positive attitude toward the purchase of electric cars, thereby increasing the purchasing intention and actual purchasing behavior of electric cars. Secondly, media propaganda, government subsidy policies, high-quality supplier services, and the construction of the Internet of Cars platform have prompted consumers to generate strong positive subjective norms, which have positively enhanced the purchasing intention and actual purchasing behavior of electric cars. Thirdly, with the economic strength and the support of charging resources, consumers have powerful perceived behavior control over the purchasing intention of electric cars, so as to exert the role of intermediate variables and promote consumers to have a hard purchasing intention and actual purchasing behavior. Finally, the fixed charging facilities, along with the susceptibility to damage and poor battery life of electric cars, play a role on consumers' purchasing intention and actual purchasing behavior. As the negative intermediate variables to consumers' attitude, the fixed attributes lowered consumers' purchasing intention and actual purchasing behavior.

The research of this paper still needs a lot of follow-up exploration. First of all, due to the concentration of sample data in Beijing, there may be certain selection biases to some extent, which make its universal promotion limited. Besides, in the process of analyzing the influencing factors of the actual purchasing behavior of electric cars, we fail to take into account specific family characteristics, such as the number of cars already owned and whether electric cars are used as the main means of transportation. There are also different brands of electric cars that have been neglected in research, such as Tesla, BMW, BYD and Toyota, which will also have diverse effects on the actual purchasing behavior of electric cars. Follow-up studies are needed to address the above deficiencies.

**Author Contributions:** Conceptualization, G.Q.; methodology, M.Z.; formal analysis, G.Q. and M.Z.; investigation, G.Q.; resources, Q.Y.; writing—original draft preparation, G.Q. and M.Z.; writing—review and editing, Q.Y., G.Q., and M.Z. The authors will be grateful for the valuable comments of reviewers from all walks of life.

**Funding:** This work is supported by the 111 project (Grant No. B18021) and 2018 Ministry of Education Key Projects of Philosophy and Social Sciences research (Grant No. JZD032).

**Acknowledgments:** We appreciate the editors and peer reviewers for their constructive comments and will reflect on the shortcomings of this paper.

**Conflicts of Interest:** The authors declare no conflicts of interest.

## Appendix A

The online questionnaires are available at https://www.wjx.cn/xz/35005235.aspx.

## Appendix B

**Table A1.** Investigative instruments.

| Construct | Items |
|---|---|
| Attitude | A1: Driving electric cars generate less pollution emissions than the conventional cars.<br>A2: Considering all costs, driving electric cars is no more expensive than driving conventional cars.<br>A3: Driving electric cars make very little noise.<br>A4: The city has issued relevant policies to support the purchase of electric cars.<br>A5: The electric cars can only be charged in fixed charging facilities.<br>A6: The electric cars often break, and their cruising range cannot meet expectation. |
| Subjective norm | S1: My influencers think I should buy an electric vehicle.<br>S2: The media push me to buy an electric vehicle.<br>S3: The government subsidies prompt me to buy an electric vehicle.<br>S4: The excellent services of the supplier prompt me to buy an electric vehicle.<br>S5: The establishment of China's internet of cars platform push me to buy an electric vehicle. |
| Perceived behavioral control | P1: I can largely decide whether to buy an electric car or not.<br>P2: I can afford to buy an electric vehicle.<br>P3: There are charging resources around my work and life to support the daily use of electric cars.<br>P4: The charging time of electric cars does not affect the daily use of electric vehicle. |
| Purchase intention | I1: Next time I buy a car, I will consider buying an electric vehicle.<br>I2: I expect to drive an electric vehicle in the near future.<br>I3: I must have an electric vehicle in the near future. |

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
