# Peer review of "Research on Real Purchasing Behavior Analysis of Electric Cars in Beijing Based on Structural Equation Modeling and Multinomial Logit Model"

_sustainability, doi:10.3390/su11205870_

Round 1

Reviewer 1 Report

Before paragraph 1.1. (row 71) necessity to develop the aim of the research and its added value

Row 73 it is affirmed that a large number of reliable data have been collected. But no proof of this is given.

As regards demographic variables, 3 variables are mentioned: Gender, income and education. The only demographic variable is gender, the 2 others are socio-economic characteristics. Why the age variable is not taken into consideration as a lot of similar researches have shown that this characteristic is often important.

There is absolutely no information about the sample size. In which extent the sample size (537 questionnaires) is relevant? How was selected the sample? These points have to be clarified.

Why a five-point likert scale was selected whereas it is generally accepted that this scale is very limited, especially when using TPB approach.

The main problems concern the presentation of the MNL results: (i) it is necessary to explain clearly what are the possible outcomes of the dependent variable, (ii) the presentation of the results are poor. Noting about Likelihood ratio tests, no presentation and interpretation of the odd ratios. No information as regards the proportional by chance accuracy rate.

Author Response

Dear Editors and Reviewers:

We would like to thank Sustainability for giving us the opportunity to revise our manuscript. Thank you for your letter and for the comments concerning our manuscript. Those comments are all valuable and very helpful for revising and improving our paper, as well as the important guiding significance to our researches. We have studied comments carefully and have made correction which we hope meet with approval. Revised portion are marked in red in the paper. The main corrections in the paper and the responds to the reviewer’s comments are as flowing.

Reviewer 2 Report

This is a well-written paper with an appropriate sample size and fine methodological background. The paper fits the Journal and especially to the SI.

Major suggestions

Line 30 - Authors write the following: “The air pollution levels in China are already above the “safe level” set by the European Union, and the primary cause is the rapid growth in the purchase and use of gasoline cars and the smaller market share of the electric vehicles.” – please add sources to these statements. Also please cite where a reader can find the “»safe level« set by the European Union”.

Authors use expression “vehicles” in the whole paper, however, the paper seems to focus only on electric cars.

Lines 37-40 - Authors write the following “According to the latest data, China sold 1.25 million electric vehicles in 2018, accounting for 4.4% of new car sales, up just 1.7 % from last year [4]. This data shows that the popularity of electric vehicles in China still has a long way to go; most consumers are in a state of wait-and-see.” – do you mean China sold inland this amount of electric cars? Is there any export that biases data?

Please add a source to the Figure 1.

In chapters 1.2., 1.3., 1.4. Authors state many statements without any citations. Are these hypotheses? It does not seem so. All of these need to have sources.

To me, the biggest concern is that Authors use a connection between “purchase intention” and “the actual purchasing power”. As they state, “Secondly, the purchasing intention is further concretized into real purchasing power to provide more favorable guidance for the diffusion policy of electric vehicles.” As this connection is not part of the original TPB, this should be introduced in more detailed. I am not sure either that purchase intention effects anyhow the purchasing power. But, purchasing power can have an effect on purchase intention. Probably, a different expression would explain better what Authors mean, or the direction of the arrow should be reversed?

Authors state in line 173 that “users have a better understanding of electric vehicles (…)” and then they state that the sample is representative. I think that the sample is very good but not representative to the Chinese population or citizens of Beijing as the respondents understand better the product than average Beijing citizens. Therefore, I think if the Authors add information, who they think the sample is representative, the statement would be more precise.

In paragraph 229-234, the Authors introduce an analysis. Please clarify the necessity of this analysis.

At the end of the paper, the Authors state that “There are also different brands of electric vehicles that have been neglected in research, such as Tesla, BMW, BYD and Toyota (...)”. It is not obvious what brands this paper selected for the research.

Table from Appendix B definitely should be in the Methodology chapter, as readers need to understand the statements have been used for this analysis.

Minor suggestions

Line 21 - the Authors write “consumer’s attitude”, but in line 22 they write “consumers’ actual purchasing power”. Decide which one to choose.

Line 67 - the Authors mention one research “the previous research (…)” but there are two references added.

Line 126 - the Authors write “consumers’ actual purchase of electric vehicles”. I think this would be better as customers’ actual purchase of electric vehicles, as customers have purchase power.

Line 190 (and Table 1) – please add currency after “monthly income of 5,000-15,000”

Line 209 – “relatively ideal” – please change the wording. Ideal or relatively good?

Line 272 – “acceptable significantly effect” – what do you mean?

Lines 278-280 – please add citation/citations

Author Response

(The authors gave the same response as above.)

Round 2

Reviewer 1 Report

I agree with the publication. All necessary corrections have been done